# Changes in the Static Balance of Older Women Participating in Regular Nordic Walking Sessions and Nordic Walking Combined with Cognitive Training

**DOI:** 10.3390/ijerph17155617

**Published:** 2020-08-04

**Authors:** Joanna Piotrowska, Monika Guszkowska, Anna Leś, Izabela Rutkowska

**Affiliations:** 1Faculty of Physical Education, Jozef Pilsudski University of Physical Education in Warsaw, 00-968 Warsaw, Poland; joanna_piotrowska@onet.eu (J.P.); les.anna@wp.pl (A.L.); 2Faculty of Rehabilitation, Jozef Pilsudski University of Physical Education in Warsaw, 00-968 Warsaw, Poland; monika.guszkowska@awf.edu.pl

**Keywords:** aging, physical activity, Nordic walking, cognitive training

## Abstract

Regular Nordic walking (NW) improves physical fitness, including the ability to maintain balance, in older adults. However, little is known about whether complementing the exercise programme with cognitive training (CT) contributes to increased effects. The aim of the study was to determine and compare the effect of regular NW and NW combined with CT on the ability to maintain static balance in older adults. The study examined 61 women aged 64 to 93 years living in adult day care centres. Twenty people participated in a three-month programme combining NW and CT (group NW + CT), 20 people participated only in NW classes (group NW), and 21 people were a control group (group C). The Romberg balance test, Fullerton Functional Fitness Test, and Attention and Perceptivity Test were used. After the programme, an increase in the time of maintaining the balance (with eyes open on the left and right legs) was observed in groups NW + CT and NW, with no such changes found in group C. This increase was greater in group NW + CT. Increased agility and strength of the hand were predictors of improving the ability to maintain balance. Regular NW improved the ability to maintain balance with eyes open in female residents of adult day care centres.

## 1. Introduction

During the ageing process, many unfavourable changes occur in the human body. Biological changes are initially imperceptible, but they lead to a gradual deterioration of all functions, including the locomotor system, internal organs, and in the psychological and social spheres [1]. The systems responsible for maintaining balance and cognitive function are weakened, and the body at this stage of ontogenesis is characterized by lower compensation and recovery capabilities [2]. Older adults experience a deterioration in all motor skills [3]. One of the major symptoms of ageing is the progressive loss of muscle weight and strength, which leads to numerous changes in body function, primarily the deterioration of physical fitness. Sarcopenia is particularly dangerous for older adults, leading to disability and mortality in this group of people [4,5]. In most older people, sarcopenia has a multifactorial background. It is a consequence of many different factors, including disturbed skeletal muscle homeostasis and, indirectly, the entire body. In the ageing body, there is then a relative advantage of catabolic over anabolic processes [6].

With age, the number of motor units gradually decreases, consequently reducing muscle function. Between 70 and 80 years of age, the number of motor units in the tibialis anterior muscle is about 50% of the value observed in young people [7]. Lower limbs are more likely to lose motor units than upper limbs, which are more resistant to progressive ageing processes [8]. Between 30 and 40 years of age, humans reach the highest level of maximum strength. As a result of involutional changes, it drops to a level of about 60–70% of the maximum values at the age of 70 [9]. Between 30 and 80 years of age, the strength of isometric contraction also decreases by about 50% [10]. The gradual decrease in muscle mass is accompanied by a decrease in muscle flexibility and, consequently, a decline in muscle efficiency. Muscle power decreases even more than muscle strength, which results from a progressive decrease in the speed of shortening of muscles with age [11]. Consequently, reaction time becomes longer, as the rate of nerve impulse conduction is reduced. This leads to limitations in walking and maintaining a stable body posture. These changes cause disorders in neuromuscular coordination and reduced ability to perform fast and rhythmic movements [12]. As a result of ageing of the locomotor system, thoracic kyphosis is increased, whereas lumbar lordosis is decreased. As a consequence of these changes, the centre of gravity moves forward with simultaneous limitation in the mobility of the limbs and spine [13].

Involutional changes in individual systems lead to substantial difficulties during walking. With larger body sway during walking and reduced step length and formation of a larger support surface (wider steps), walking becomes less and less economic, leading to greater fatigue, turning into an “old age gait” [14]. This slower gait is strongly connected with the weakening of cognitive function, and at the same time, it represents a predictor of cognitive dysfunction [15]. Deterioration of the ability to maintain balance, which is one of the coordination capacities, reduces movement control and consequently increases the risk of falling. Balance disorders and the related falls are associated with injury. They can cause fear of movement and lead to loss of motor independence [16].

As the number of older adults is increasing every year, it is advisable to look for factors influencing the best possible psychophysical status of the person in the longest possible time [17]. It is becoming necessary to develop and implement special prevention programmes aimed at maintaining and even improving the fitness of the elderly, especially in terms of balance and muscle strength. The results of Nordic walking (NW) classes appear to be promising. The female participants of the Third Age University programme who attended regular NW classes for three months improved their results in functional fitness tests, especially in tests that measured physical capacity and strength [18]. Ossowski and Kortas [19], after completion of a six-month NW programme, documented improvements in balance and agility of older adults. In the study by Piotrowska and her team [20], older people participating in a 12-week programme of NW exercises and cognitive training (CT), improved their performance both in terms of balance and functional/cognitive fitness.

The results of previous research indicate that regular exercise, such as NW, improves physical fitness, including the ability to maintain balance, in older adults. However, little is known about whether complementing the exercise programme with CT contributes to increased effects. There are more and more reports in the world literature that emphasize the role of cognitive functions as components of motor control. Studies confirmed that the speed of walking after adding a cognitive task correlates with the efficiency of executive functions [21]. The research confirmed that the walking speed of the elderly, which is a manifestation of the efficiency of executive functions, significantly decreased after adding the cognitive task.

In the ageing process, executive functions are weakened, which affects the functional efficiency of seniors. Additionally, weakening of the muscular system function translates into impairment of body posture control function and an increased risk of falls [22,23]. Cognitive processes play an important role in the control of motor functions, and their improvement may cause an increase in the functional efficiency of seniors [24]. The combination of NW and CT was aimed at checking whether this combined intervention would have a greater impact on the ability of the elderly to maintain balance.

The question also arises as to what the changes in balance depend on and which factors make it possible to anticipate them. The aim of the study was to determine and compare the effect of regular NW exercises and NW exercises combined with CT on the ability to maintain balance in older women. We assumed that the effects in group NW + CT would be greater than in group NW. Relationships between changes in ability to maintain balance and changes in physical and perceptual fitness were also examined.

## 2. Materials and Methods

### 2.1. Research Design

The examinations were performed using the experimental method in natural conditions. The participants were residents of three elderly day care centres in Warsaw. Three groups were formed: Two experimental groups—the group that participated in NW sessions and CT (group NW + CT) and the group participating only in NW sessions (group NW)—and the control group without interventions (group C). Each group included residents from different day care centres. Purposive sampling with “threes” was employed to ensure evenness of the groups while adopting the criterion of general health status and physical and cognitive fitness.

NW exercises were aimed at improving aerobic capacity, joint stabilization, flexibility, increasing muscle strength and improving static and dynamic balance and motor coordination (improvement of functional fitness). The programme took three months, with training sessions held twice a week for 60 min. Classes included a 10-min warm up, exercises to improve muscle strength (15 min) and breathing and flexibility exercises, with a duration of 5 min. During the classes, participants walked for 30 min and covered an average distance of 2.5 km (min. 1.5 km, max. 3.2 km). During the first 5 min (5–8 weeks) and 10 min (9–12 weeks) of the walk, the participants from the groups NW and NW + CT performed additional tasks, such as solving arithmetic problems (e.g., subtraction every 3 from 197). During these exercises, the walking speed was definitely decreased by 2 km/h on average.

During each class, attention was paid to maintain the correct technique of walking with poles, particularly the pole planting and push phase. The participants from both experimental groups NW + CT and NW covered a greater number of kilometres each month. The route was measured using dedicated equipment with an in-built GPS POLAR m400 module operated by a coach who supervised training.

The author’s programme of exercises of selected cognitive functions was used in the CT in accordance with the multicomponent Baddeley’s model of working memory [25]. It included exercises aimed to improve coordinated functions of attention and working memory, training using the visuospatial sketchpad and exercises of management functions. The training was complemented by increasing the working memory capacity using the mnemotechnic chain method (associating words in specific order) and symbol method (consisting of associating words with a number having its equivalent in the form of a symbol; this method allows the person to memorize and then to retrieve the elements in any order). The programme took three months, with training sessions held twice a week for 60 min.

The group without interventions participated in typical classes offered to the residents of the day care centres.

### 2.2. Sample

A total of 88 residents aged over 60 years participated in the experiment. The inclusion criterion was at least average (for the appropriate age group) level of health, physical fitness and basic cognitive functions. The data on health condition of the respondents was obtained from the physician caring for residents. The participants were elderly people without some kinds of serious health problems, namely cardiovascular and locomotor system diseases, and neurological diseases. The data on physical fitness was obtained from caregivers in day care centres. Furthermore, the participant had to be able to walk 1.5 km without pain or breathing problems. Additionally, we applied the Katz scale (ADL—Activities of Daily Living) and the Lawton scale (IADL—Instrumental Activities of Daily Living) to determine whether the subjects could perform their daily activities independently, safely and without too much effort. Information on the cognitive possibilities was provided by a psychologist or a geriatrician. All the subjects were independent in terms of their motor skills and performed their daily activities independently. Furthermore, we used the Minimental Test—Mini Mental State Examination (MMSE). The participants had to achieve the maximum result in this test to be qualified to our study.

People suffering from chronic diseases of the cardiovascular system, locomotor system and neurological diseases that could affect the results of the balance and physical fitness tests, and people with limited cognitive abilities (dementia) were excluded from the study.

The laterality of the limbs was assessed by checking which hand the subject used while writing and asking her/him to kick a small ball. People with homogeneous right laterality participated in the study.

An additional criterion of inclusion into analyses in the case of the experimental groups was participation in at least 85% of sessions. The final analyses involved 68 people who participated in all measurements of physical parameters and physical and cognitive fitness. Due to an insignificant number of men (*n* = 7) meeting the inclusion criteria, the study presents the data concerning only 61 women.

The age of participants ranged from 64 to 93 years (M = 80.25; SD = 5.755). They were mostly widows (*n* = 54) and women living alone (*n* = 51). Women included in the study most often had secondary education (*n* = 47) and vocational education (*n* = 9). Four participants had primary education, and one participant had higher education. All the participants had been employed in the past and had performed physical work (*n* = 40) more often than mental (*n* = 11) or mixed (*n* = 10) work. The study participants evaluated their financial status mostly as average (*n* = 41) and less often as good (*n* = 11) and poor (*n* = 7). The majority of the women assessed their former lifestyles as medium active (*n* = 35) and less often as active (*n* = 17) and very active (*n* = 8); 32 participants were very rarely involved in physical activity, and 22 of them did this at any opportunity. At the time of the study, the respondents travelled mainly by public transportation (*n* = 46) and less frequently by car (*n* = 9) or on foot (*n* = 5). Furthermore, 37 respondents declared that, in the past, they did not pay attention to healthy diets, whereas the others answered in the affirmative.

The groups NW + CT and NW had 20 people each, whereas group C consisted of 21 participants. The groups did not differ significantly in terms of age (F = 0.528; *p* = 0.592), education (chi squared = 4.723; *p* = 0.580), marital status (chi squared = 7.036; *p* = 0.318), living alone/with family (chi squared = 2.766; *p* = 0.598) and economic status (chi squared = 9.669; *p* = 0.139). The groups were evenly matched in terms of baseline physical fitness and the ability to maintain balance.

### 2.3. Data Collection

Data on age, marital status, education, professional activity, financial status and physical activity in the past and currently were obtained by means of an author’s survey that was used in previous studies of the elderly [18,20].

In order to evaluate the ability to maintain balance, the Romberg balance test was performed with hands along the body or crossed on the chest on the left and right leg with eyes open and closed [26].

The Fullerton Fitness Test was performed to evaluate physical fitness. The test is composed of six consecutive trials that evaluate [27]:Arm Curl. Evaluation of upper body strength.30-s Chair Stand. Evaluation of lower body strength.Back Scratch Test. Examination of upper body mobility.Chair Sit and Reach Test. The aim of the test is to assess the flexibility of the lower body (especially the popliteus tendon).8-Foot Up-and-Go and 6-Min Walk tests. Evaluation of agility (dynamic balance) and long-distance aerobic endurance.2-Min Step in Place Test. Evaluation of aerobic endurance is performed if the 6-Min Walk test is impossible.

Furthermore, the hand grip test was performed using a dynamometer. Each of the elderly participants, after hearing the instructions, squeezed the dynamometer bar twice with both the right and left hands in a sitting position. The result of the second test was adopted as an indicator of hand strength.

In order to determine the perceptual efficiency, the Attention and Perceptivity Test was used [28], which consists of crossing the indicated digits and provides indices of perceptual work rate (number of characters analysed), perception fallibility (number of mistakes) and attention fallibility (number of skipping instances). Standard conditions were used when performing the test (duration of 3 min). The tests were conducted in small groups of several people. Before the test began, it was verified whether each participant understood the instructions correctly.

All measurements were carried out twice: Before and after the NW programme.

The study was conducted according to the ethical guidelines and principles of the Declaration of Helsinki. All subjects gave their informed written consent to the experimental procedures, which were approved by the Senate Ethics Committee of Scientific Research AWF Warsaw (SKE 01-46/2016).

### 2.4. Statistical Analysis

Statistical analyses were performed using IBM SPSS version 25. (Armonk, NY, USA) In order to determine the normality of variable distributions, the Kolmogorov–Smirnov test was used. Repeated measures analysis of variance (group x measurement: 3 × 2) was used to determine intergroup differences and changes in time. One-way analysis of variance (ANOVA) and Tukey’s post-hoc tests were used to determine the differences between the groups. The study also examined the relationships of changes in the ability to maintain body balance with age and changes in functional and cognitive fitness. In order to evaluate these relationships, the first step was to compute the indices of changes in functional and cognitive fitness (perceptual work rate and perception and attention fallibility), then Pearson’s r linear correlation coefficients were calculated for the entire study group. The stepwise regression analysis was used to determine which indices of change made it possible to anticipate an improvement in the ability to maintain balance with eyes open and closed.

## 3. Results

### 3.1. Changes in the Ability to Maintain Balance

Repeated measures ANOVA was used to determine intergroup differences and changes in time. The respective data are presented in Table 1 and Table 2. The main effect of the group was not significant in any case. The main effect of the measurement was significant for the results of the tests with eyes open, both for standing on the right and left legs. The results of the second measurement were better compared to the first. The interaction effect was significant in three cases (except for the test with eyes closed on the right leg). In the test with eyes open on the left and right legs, the improvement of results was found only in the experimental groups (groups NW and NW + CT). No significant intergroup differences were found for these tests (in the first or second measurement) (Table 1). In the test with eyes closed, the post-hoc tests indicated no intergroup differences in the first and second measurements and no differences between measurements in all groups (Table 2).

The differences in the ability to maintain body balance were also computed by subtracting the first measurement from the result of the second measurement, where a positive value means an increase in the time of maintaining body balance. One-way ANOVA and Tukey’s post-hoc tests were used to determine the significance of differences between the groups (Table 3).

The groups differed significantly in terms of changes in the ability to maintain body balance with eyes open and closed standing on the left leg. In group C, all change indices were negative, which indicates a reduction in execution time. In group NW, the difference was positive in the tests with eyes open, whereas during the test with eyes closed, this value was close to 0 for the left leg and negative in the test of standing on the right leg. In group NW + CT, the difference was positive in all tests. Improvement indices in tests with eyes open were the highest in group NW + CT and lowest in group C. In the test of standing on the left leg with eyes closed, the improvement was significantly greater in group NW + CT than in group C. No significant intergroup differences were found in the test of standing on the right leg with eyes closed (Table 3).

### 3.2. Correlations between the Improvement in the Ability to Maintain Balance and Improvement in Physical and Cognitive Fitness

The study also examined the relationships for changes in ability to maintain body balance with age and changes in physical and cognitive fitness. In order to evaluate these relationships, the first step was to compute the differences in physical and cognitive fitness (perceptual work rate and perception and attention fallibility), then Pearson’s r linear correlation coefficients were calculated for the entire study group.

The improvement in the ability to maintain balance did not correlate significantly with the age of respondents. The correlation between the improvement in the test of balance on the right leg with eyes closed reached a trend level (r = 0.236; *p* = 0.067) and had a positive value (which may be surprising). Numerous positive relationships were also found with indices of improvement in physical and perceptual fitness (Table 4). Interestingly, most of them were found for the improvement in the ability to maintain balance on the left leg with eyes open. It was directly proportional to almost all indices of improvement of physical fitness (except for the improvement in mobility of the upper body in the left-hand test). Slightly fewer relationships were found in the case of improving the ability to maintain balance on the right leg with eyes open. Both indices correlated positively with all indices of improvement in perceptual efficiency.

The improvement in balance on the left leg with eyes closed correlated significantly with the increase in the strength of the lower body, the strength of both hands and the increase in aerobic endurance. Correlations with the increase in agility and perception speed reached only a trend level. Interestingly, no significant correlations were found for the index of improvement in the ability to maintain balance on the right leg with eyes closed.

### 3.3. Predictors of Improving the Ability to Maintain Balance

The next step was to determine which indices of change made it possible to anticipate an improvement in the ability to maintain balance with eyes open and closed. The stepwise regression analysis was used for this purpose. The factors (explanatory variables) were changes in physical parameters and in physical and cognitive fitness, whereas the dependent (explained) variables were total indices of improvement in the ability to maintain balance with eyes open and closed. These indices were the sum of the indices of improvement in the ability to maintain balance in two trials (on the right and left leg). The results of the analyses are shown in Table 5. The predictors of the improvement in balance with eyes open were the increase in agility and increase in the strength of the right hand, which, in total, allowed for prediction of the improvement in the ability to maintain balance with eyes open in over 40%. The increase in left hand strength allowed for prediction of an improvement in balance with eyes closed only in 7% of participants.

With the introduction of an additional factor (group affiliation), this was the only predictor of improvement in the ability to maintain balance with eyes closed (R^2^ = 0.478; F = 55.892; *p* < 0.001; beta = 0.697). In this case, none of the factors allowed for prediction of the improvement in the ability to maintain balance with closed eyes.

## 4. Discussion

Functional efficiency deteriorates with age. This is particularly important for people who do not engage in any physical activity for various reasons. Balance disturbances and problems with proper gait may result in serious injury and lead to fear of falling [29]. The results of our study indicate that systematic NW exercises improve the static balance of older women. We can expect that those participating in NW training will be at a lower risk of falling. This will reduce the fear of independent movement and risk of losing independence in everyday life. The World Health Organization (WHO) recommends that older people take up physical activity on a regular basis and, if possible, every day. NW is a safe form of physical activity for every human being, since it is based on cyclic and simple movements.

However, the improvement of static balance took place in both experimental groups only in the test with the eyes open. A study by Melzer et al. [30] found that a much higher concentration is needed in older people, compared to young people, to perform balance tests if memory tasks are additionally performed. The response to body sway in order to maintain balance is limited and requires reorganisation in the system of sensorimotor control. During equilibrium tests with eyes open and closed, older people stiffen the muscles of the lower limbs. The contribution of visual control in maintaining static balance is smaller than in the case of young people. The most important element of the sensory system that determines postural stability is the function of sensory receptors [31]. The NW exercises activate both the vision analyser and proprioception. In the absence of vision, complete concentration on the proprioceptive information is observed. This may be the cause of no improvements in the results of the Romberg tests with eyes closed.

Improvements in the ability to maintain balance with eyes open were found in both experimental groups. The subjects from group NW participated only in regular NW classes, performing simple cognitive exercises during the walk. People from group NW + CT performed additional memory training. However, they did not differ significantly in terms of the results of the balance test in the first and second measurements. The only significant difference was in the ability to maintain balance on the right leg with eyes open. A greater improvement in results was observed in group NW + CT. However, the groups differed significantly in terms of the indices of changes in the ability to maintain body balance with eyes opened. These results suggest that combining NW with simple exercises involving cognitive functions yields results only slightly worse than the programme composed of NW and cognitive (memory) training. Our hypothesis was therefore only partially confirmed.

The degree of cognitive involvement in NW depends on additional stimuli occurring during the walk (e.g., on uneven and unstable ground). It requires performing sudden movements or re-establishing previously learned movement patterns, which may cause problems for older adults due to lower cognitive skills [32]. In our experiment, the study participants from groups NW and NW + CT performed simple exercises involving cognitive functions. Westlake et al. [33] found that performing a sufficiently difficult additional task often leads to a reduction in problems with balance. Focusing on a secondary task leads to increased stability through a hidden learning effect. It was also observed that during the performance of the so-called double task, the walking speed increased, whereas the risk of falls decreased [34]. This effect was also observed in the women analysed in our study. Therefore, it seems necessary to use not only physical exercises in programmes of activation for older adults but also to improve their cognitive functions.

Grip strength was a predictor of the improvement in balance, both with the eyes open and closed. Garcia et al. [35] demonstrated a relationship between the grip force and walking speed. Beseler et al. [36] studied the relationship between the strength of lower limb muscles, especially those responsible for correct gait, and grip strength in older adults requiring hospitalization. They found a relationship between grip strength and mobility resulting from the strength of lower limb muscles. It was also shown that grip strength is related to cognitive disorders.

In the study, older people from groups NW and NW + CT achieved improved results in the “8-Foot Up-and-Go” test (agility). It was a predictor of the improvement in maintaining balance with eyes open. Getting up from a chair depends on the level of strength of the lower limb muscles and the ability to maintain balance [37]. Repeating this activity increases the strength of the lower limb muscles and contributes to an increase in walking speed [38]. Elderly people had the greatest problems in maintaining balance during the shift from the sitting to standing position [39]. Therefore, there are mutual relationships between the ability to maintain balance and agility, with both abilities determining gait speed, which also increased in our study participants.

During the 12-week NW programme, the participants in our experiment improved their walking by the fact that they covered more and more kilometres during their 30-min sessions. It is very important to use the correct walking technique, especially the grip technique, planting the poles in the ground and the push phase. This increases the involvement of the upper body and, consequently, the walking speed [40]. Proper use of poles helps shift the centre of gravity backward, thus allowing the person to restore the correct gait pattern and adopt the correct body posture. Such effects have been documented in studies of healthy adults [41,42,43]. It was shown that even a six-week NW training programme increased walking speed and step length in older people [44,45]. After five months of NW training, a greater range of mobility was observed in plantar- and dorsiflexion of the foot, which is of great importance for correct gait. Furthermore, the women studied had a greater range of mobility in their hip joints, a greater level of muscle strength in the elbow joints and improved dynamic balance [46].

The presented research confirms the need for developing and implementing special programmes aimed at older people. Ostrowska et al. [47], after analysis of the gait of older adults, recommended that the programme for such people should include balance training and exercises improving mobility in the hip and knee joints and strengthening the erector spinae muscles, which increases joint mobility. Maintaining an adequate level of physical activity in older adults is important not only for their physical fitness but also for the associated cognitive functions related to reception and selection of information coming from the environment. Programming physical activity for older adults requires a comprehensive and individual approach depending on the level of physical and cognitive performance. Regular NW classes and CT can offer an excellent tool for the development of physical fitness and cognitive function in older adults. The exercise programme should be clearly defined in terms of intensity and frequency in order to improve adaptability to changes that occur with age. The attractiveness of the programme is also important for motivating older people. The NW programme adjusted for older adults and supervised by coaches can be an effective strategy to improve functional fitness and, consequently, to delay the period of disability. Motivating older adults for undertaking physical activity is particularly important in day care centres that help people with lower physical and cognitive fitness. In future studies, authors should determine and compare the effect of regular NW combined with CT on the ability to maintain static and also dynamic balance in older adults with some physical or/and mental diseases.

### Limitations

The results of our study are not free of limitations. The first limitation is related to the experimental design used in our study. For organizational reasons, it was possible to create only three study groups. A full experimental model would require the creation of an additional group participating only in CT. This would better distinguish between the effects of NW training and CT.

The second limitation results from the choice of the respondents. The groups were recruited from residents of different day care centres. This solution was chosen to avoid a situation where only some residents would participate in additional activities. We attempted to limit the effect of this factor by making the recruitment in “threes.” However, it cannot be excluded that differences in the social environment had, at least in part, an impact on the results obtained.

With the insignificant percentage of men in the centres, we analysed only the results obtained for women, and the relationships found relate only to this sex. The possibility of generalization of the results is additionally limited by the fact that our participants were quite healthy and physically and cognitively fit for their age. Moreover, they lived in the capital city. This group is not representative for the entire population of older adults in Poland. In the capital city, more opportunities are offered for older adults spending their leisure time actively compared to rural areas.

The authors are aware that the number of subjects did not provide adequate testing power. The initial calculations at the research planning stage assumed larger differences and slightly less variance of results. It should be noted, however, that the study was conducted in a specific population and the intervention lasted a relatively long time. Gathering 60 people and conducting research as intended was a challenge. Every effort was made to ensure that the study was conducted accurately. However, the small number of studied groups remains a limitation.

Another limitation concerns the CT and cognitive function indices used in the study. Further research is needed to compare the effects of training focused on different cognitive functions. It is necessary to use tools that allow for their reliable and accurate measurement.

## 5. Conclusions

Implementation of specialized programmes with the use of NW poles for Social Welfare Homes can be an effective strategy to maintain and improve the ability to maintain balance with eyes open in elderly people with at least average health condition as well as physical and cognitive fitness. It seems necessary to introduce an additional element, CT, to the programmes for healthy and fit elderly people. The inclusion of elements of CT increased, although slightly, benefits in the ability to maintain balance with eyes open, especially when standing on the right leg. In elderly people who did not participate in motor activities, there were no changes in balance in short-term observations.

## Figures and Tables

**Table 1 ijerph-17-05617-t001:** The results of the Romberg test for measuring the ability to maintain balance (comparison by group and measurement).

Test	Measurement		Group		ANOVA			
	First (M ± SD)	Second (M ± SD)	C (M *)	NW (M *)	NW + CT (M *)	Measurement (F, *p*, η^2^)	Group (F, *p*, η^2^)	Interaction (F, *p*, η^2^)
With eyes open, on left leg ^1^	3.89 ± 6.92	4.84 ± 6.86	3.74	4.65	4.72	26.24; <0.001; 0.311	0.130, *ns*	22.43; <0.001; 0.436
With eyes open, on right leg ^1^	3.79 ± 6.41	4.89 ± 6.66	2.93	5.22	4.92	42.01; <0.001; 0.420	0.761; *ns*	21.86; <0.001; 0.430
With eyes closed, on left leg ^1^	1.62 ± 2.96	1.62 ± 3.08	0.88	2.37	1.65	0.003; *ns*	1.28; *ns*	3.29; 0.044; 0.102
With eyes closed, on right leg ^1^	1.87 ± 3.27	1.79 ± 3.07	1.33	2.50	1.67	0.740; *ns*	0.366; *ns*	0.282; *ns*

^1^—measured in seconds; the longer the time, the better the ability to maintain balance; M *—weighted averages; *ns*—not statistically significant.

**Table 2 ijerph-17-05617-t002:** The results of the Romberg test for measuring the ability to maintain balance in subgroups (Tukey’s post-hoc test).

Test	Measurement	Group C (1) (M ± SD)	Comparison of Measurements *p* *	Group NW (2) (M ± SD)	Comparison of Measurements *p* *	Group NW + CT (3) (M ± SD)	Comparison of Measurement *p* *	Comparison of Groups		
								1–2 *p* *	1–3 *p* *	2–3 *p* *
With eyes open, on left leg ^1^	I	4.10 ± 7.327	*ns*	4.00 ± 6.681	0.003	3.55 ± 7.060	<0.001	*ns*	*ns*	*ns*
	II	3.38 ± 7.22		5.30 ± 6.72		5.90 ± 6.70		*ns*	*ns*	*ns*
With yes open, on right leg ^1^	I	3.05 ± 3.92	*ns*	4.70 ± 7.85	0.012	3.65 ± 7.10	<0.001	*ns*	*ns*	*ns*
	II	2.81 ± 3.92		5.75 ± 8.51		6.20 ± 6.62		*ns*	*ns*	*ns*
With eyes closed, on left leg ^1^	I	1.00 ± 1.26	*ns*	2.35 ± 3.96	*ns*	1.55 ± 3.03	*ns*	*ns*	*ns*	*ns*
	II	0.76 ± 1.26		2.40 ± 4.17		1.75 ± 3.06		*ns*	*ns*	*ns*
With eyes closed, on right leg ^1^	I	1.43 ± 1.66	*ns*	2.55 ± 4.41	*ns*	1.65 ± 3.26	*ns*	*ns*	*ns*	*ns*
	II	1.24 ± 1.55		2.45 ± 4.15		1.70 ± 3.04		*ns*	*ns*	*ns*

^1^—measured in seconds; the longer the time, the better the ability to maintain balance; *—Tukey’s post-hoc test; *ns*—not statistically significant.

**Table 3 ijerph-17-05617-t003:** Changes in the results of the Romberg test for measuring the ability to maintain balance in subgroups (ANOVA).

Test	Group			ANOVA (F, *p*, η^2^)	Post-hoc Tests		
	C (1) (M ± SD)	NW (2) (M ± SD)	NW + CT (3) (M ± SD)		1–2	1–3	2–3
With eyes open, on left leg ^1^	−0.71 ± 0.72	1.30 ± 1.56	2.35 ± 1.95	22.43; <0.001; 0.436	<0.001	<0.001	0.075
With eyes open, on right leg ^1^	−0.24 ± 0.54	1.05 ± 1.15	2.55 ± 1.99	21.86; <0.001; 0.430	0.009	<0.001	0.002
With eyes closed, on left leg ^1^	−0.24 ± 0.62	0.05 ± 0.51	0.20 ± 0.52	3.29; 0.044; *ns*	*ns*	0.038	*ns*
With eyes closed, on right leg ^1^	−0.19 ± 1.33	−0.10 ± 0.97	0.05 ± 0.69	0.282; *ns*	*ns*	*ns*	*ns*

^1^—measured in seconds; the longer the time, the better the ability to maintain balance; *ns*—not statistically significant.

**Table 4 ijerph-17-05617-t004:** Correlations between the improvement in the ability to maintain balance and improvement in physical and cognitive fitness (Pearson’s r coefficients).

Area	Variable	Improved Balance (with Eyes Open, on Left Leg)	Improved Balance (with Eyes Open, on Right Leg)	Improved Balance (with Eyes Closed, on Left Leg)	Improved Balance (with Eyes Closed, on Right Leg)
Physical fitness	Increased strength of the left hand (dynamometer)	0.312 *	0.189	0.032	0.051
	Increased strength of the right hand (dynamometer)	0.451 **	0.277 *	0.086	0.134
	Increased aerobic endurance ^1^	0.446 **	0.374 **	0.267 *	0.109
	Improved flexibility (left leg) ^2^	0.299 *	0.228 ^t^	0.118	−0.011
	Improved flexibility (right leg) ^2^	0.295 *	0.196	0.185	0.072
	Improved upper body mobility (left limb) ^3^	0.073	0.075	0.016	−0.018
	Improved upper body mobility (right limb) ^3^	0.350 **	0.318 *	0.149	−0.091
	Increased lower body strength ^4^	0.560 **	0.451 **	0.324 **	0.093
	Increased agility ^5^	0.584 **	0.533 **	0.250 ^t^	0.135
	Increased upper body^6^ strength (left hand)	0.428 **	0.404 **	0.321 *	0.212
	Increased upper body strength (right hand) ^6^	0.524 **	0.576 **	0.323 *	0.140
Perceptual efficiency	Improved perception speed ^7^	0.406 **	0.407 **	0.234 ^t^	0.100
	Decrease in perception fallibility ^8^	0.392 **	0.300 *	0.211	0.085

^1^ 6-Min Walk (m); ^2^ Chair Sit and Reach Test (cm); ^3^ Back Scratch Test (cm); ^4^ 30-s Chair Stand (the number of stands); ^5^ 8-Foot Up and Go (sec); ^6^ Arm Curl (the number of lifts); ^7^ number of characters analysed; ^8^ number of mistakes; ** *p* < 0.001; * *p* < 0.05; ^t^
*p* < 0.1.

**Table 5 ijerph-17-05617-t005:** Predictors of improving the ability to maintain balance with eyes open and closed (stepwise regression analysis).

Dependent Variable	Step	Predictor	Beta	Model (R^2^; F; *p*)
Improved balance—with eyes open	1.	Increased agility ^2^	0.599	0.347; 32.943, <0.001
	2.	Increased agility ^2^	0.390	0.426; 23.240; <0.001
		Increased strength of the right hand ^3^	0.361	
Improved balance—with eyes closed	1.	Increased strength of the left hand ^3^	0.293	0.071; 5.553; 0.022

^1^ (sec), a positive value indicates an improvement in the result; ^2^ 8-Foot Up and Go (sec), a positive value indicates an improvement in the result; ^3^ dynamometer; a positive value indicates an improvement in the result.

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
