# Peer review of "Changes in the Static Balance of Older Women Participating in Regular Nordic Walking Sessions and Nordic Walking Combined with Cognitive Training"

_ijerph, 2020, doi:10.3390/ijerph17155617_

Round 1

Reviewer 1 Report

The authors have addressed nearly all of my previous comments and have definitely improved the manuscript.

Minor comments 

  1. Can the authors clearly define if the sample size was adequately powered or similar to previous studies and if not, could you include this within the limitations section. 
  2. Can the authors define how parametricity was determined in the statistical analysis section.

Reviewer 2 Report

My previous comments have been addressed although I did note

L 382-384 should be deleted as these were instructions for authors.

L 363 should be plural; limitations

L 271 table is split over two pages

L 192 should be on the next page

Reviewer 3 Report

This intervention study reports the effects of Nordic walking and of Nordic walking combined with cognitive training on the static balance of older women. The paper is well-focused on the topic and reports results with practical relevance. Introduction provides sufficient background with relevant references. The research design enables studying the set research questions. The report section is especially well-organized with informative sub-titles. The paper is well-written and has good flow in text. However, there are some issues, which I see necessary to be clarified before I would recommend this paper to be published.

Did you have information on chronic conditions of the participants? E.g. neurological diseases may affect balance as well as physical and cognitive capabilities, including achievable training effects. Therefore, in studying intervention effects on balance, it would be important to exclude the effect of participants’ health conditions on balance and on changes in it. If possible, please provide more information on how health conditions were considered in participant selection and in assigning them to groups and, if relevant, how the health conditions may have affected the results.

Most people have either left or right foot / hand dominant over the other, which may possibly manifest as higher performance levels in e.g. balance and hand grip strength in the dominant foot / hand. Do you think dominant foot or dominant hand plays any role in how individuals are able to improve their performance? The role of dominant foot / hand is currently not discussed in the paper.

For the predictor variables, please provide characteristics data and information on the representativeness of the study sample in relation to wider population. Has this group of older women the same physical and cognitive capabilities as the population in this age group in general? This information would help understanding how representative the results may be at population level and to whom this type of intervention might provide benefits.

Please provide units for the measured items and inform whether a higher value indicates better capability. This would help the reader to interpret the analyses results (especially the regression analyses).

Despite of the paper being well-written in general, but I suggest paying special attention to making for the reader clear, which groups are compared in each comparison. This applies to the table titles as well. Table titles could help the reader to orient for retrieving relevant information from the table. Currently the titles do not succeed in this. E.g. table 1 title “Comparison of the results of the test of ability to maintain balance …”. does not indicate between which / what the comparison is made (is it made between individuals within groups, groups, measurements, or what) and the term indicating the measurement done (test of ability to maintain balance) in the title does not match with the term used in the table (eyes open, …).  

I suggest considering again the term used for reporting changes in the ability to maintain balance. If I understood correctly, currently the authors refer with the term “index” to the plain difference between measurement II result minus measurement I result, thus the index showing difference between the measurement results in seconds. As there is e.g. no scaling done to show relative difference between the measurement results, I think the term “difference” would be more appropriate instead of “index”.

Table 1 reports weighted averages for the balance tests at group level. Please explain how these figures are formed, what they tell, and for what purpose they are used. Currently the functions of these figures remain unclear.

Please provide information on how the “total indices of improvement in the ability to maintain balance with eyes open and closed” (line 265) was formed?

For the tables, please provide notes below the table to explain the abbreviations used in the tables.

Please check table layouts. In the tables there are e.g. unnecessary horizontal lines and inconsistency in indicating statistically significant results in bold. In table 4, one of the correlation coefficients of Increased lower body strength (Chair) is one row upper than the others.

Which statistical software program did you use in the analyses?

Reviewer 4 Report

The study is relevant, it was well conducted and presents important contributions to scientific literature.

However, the authors could explore further on state of the art. As well as making it clearer which research question the study intends to answer. It is also important to make clear what the practical implications and direction for future studies.

There are a lot of references in Polish. It is necessary to use references that readers can access.

Author Response

This manuscript is a resubmission of an earlier submission. The following is a list of the peer review reports and author responses from that submission.

Round 1

Reviewer 1 Report

The authors present an interesting article demonstrating the beneficial effects of Nordic walking in older adults on postural balance in females aged between 64 and 93 years old. Whilst there were no differences reported between Nordic walking training and combined Nordic walking training and cognitive training, additional benefits were reported, yet not significant.

Abstract

Line 14 NW needs defining as it is its first use.

Line 14-15: Can the first sentence be rewritten to be more concise. I would suggest that the first six six words can be removed.

Can the authors present their data in the abstract to detail any changes that were present.

Were any differences observed between Nordic Walking groups?

What is the clear conclusion of the study, which is supported by the results?

Introduction

Line 36-38: Can the authors classify what they mean by weight loss? Do they mean muscle in reference to sarcopenia? In addition what are the mechanisms that create this, as it is not only due to ageing.

Line 73: change older persons to older adults

Line 49: remove the word ‘the’ after consequently

Can the authors provide a rationale to why they believe supplementing memory training in their intervention will increase balance? This is because it is one of the main focuses of the research, but it is missing within the introduction.

Materials and Methods

Were the groups adequately powered? Can this be added to the manuscript?

Can the authors elaborate on the method they undertook to assess handgrip strength.

Were baseline fitness levels controlled for, when undertaking the Repeated measures ANOVA?

Were physical levels screened prior to the study and were trained participants excluded?

Results

Can the authors divide the section into subsections to make the results clearer to the reader?

Can the authors make it clear what table does the data they are discussing appears in.

Line 206: What post hoc tests were undertaken?

Table 1 & Table 2: Can the use of decimal places be consistent

Discussion

 Whilst it is clear from the results on the benefits of physical activity in older adults, can this be amalgamated within the opening paragraph of the discussion?

Limitations

I believe in this section the authors should state that four groups should have been examined, with the missing group only to have undertaken cognitive training.

Conclusion

Whilst additional benefits may be acquired from older adults with whom who undertake both Nordic Walking and cognitive training, these were not significantly different from participants whom only undertook Nordic walking training. Therefore, this should be reflected and made more clearer within the conclusion.

Reviewer 2 Report

A brief summary

The authors present a study where the effects of Nordic walking together with cognitive training on the older adults’ ability to maintain static balance are observed. They proposed a pre-post test with two experimental groups and a control group. Test assessing the static balance and physical fitness are carried out finding improvements in the experimental groups’ balance without between-group differences.

Broad comments

The study objective is of great interest to today's society. Aging is increasing at a rapid rate and the number of old people is getting higher and higher. It is necessary to find strategies to guarantee the quality of life of this society’s strata. In my opinion, the study design is correct, but I have some concerns with the randomization.

The abstract and lines 76-77 indicate the previous evidence of the beneficial effects of exercise in the ability to maintain balance. Which way do you think the results provide an advance in current knowledge? By the way, why three groups? Is one of the objectives the determination of the cognitive training effects by its own? Why does not the title reflect this? Why this objective not commented in the discussion or the conclusions?

There are some variables (like arm curl test and Back Scratch test) which are not reported in the results and commented in the discussion. Besides, the authors mention a survey “developed specifically for the research”. Is it validated? there are no validated surveys to report this data?

The statistical method performed is problematic. The one-way ANOVA allows the analysis of one moment. This study has pre-post test, so the correct statistical method would have been ANOVA 3x2 analyzing three groups in two moments.

Specific comments

 You need to define your abbreviations the first time they appear (line 14) then, use only the abbreviation (lines 69, 100, 101, 120, 151, 279, 292, 345, 373). Please, eliminate the second "increased" to improve the fluency of the sentence. Please, spell correctly “aging” not “ageing” (line 31). A reference is needed for these affirmations (lines 48-52). In the sentence “reduced step length and formation of a larger support surface (wider steps)” there could be opposed information (Lines 57-58).  I am not sure I understand the meaning of this sentence “later cause fear of loss of independence” could you please modify it to make it clearer? Please, add the reference for this research (line 76-77). Before submitting a paper it should be previously reviewed carefully. Please, delete this paragraph belonging to the instructions template (lines 85-94). Is not necessary to explain the research design each time that you mention the group names. Please, just say: NW+C and NW group had 20 subjects each (lines 151-152).

Reviewer 3 Report

This is a good study that does shed light on the interaction between nordic walking and cognitive exercises on balance. There are some minor edits needed such as:

  • title does not reflect the cognitive intervention
  • abstract does not spell our NW in the first instance
  • Both the methods and results are prefaced by instructions for the authors which should be deleted.
  • Results tables should indicate the unit of measurement

The introduction provides adequate background information, methods are clear and design is appropriate, discussion/conclusions are logical, limitations are acknowledged.

Reviewer 4 Report

This paper addresses an interesting topic and examine a novel intervention. The paper's presentation, however poses significant issues for publication. Issues: 1. There are a large number of awkward phrases that makes it very difficult to understand. 2. The small numbers should be listed as a limitation. 3. The introduction should be shorter. 4. The methods state that there are 2 groups, then lists 3 which is confusing. 5. Was the exercise/cognitive exercise group called NW + C? or NW + P? 6. The results/tables state numerous non-significant results but does not provide P values. 7. The beginning of the discussion needs to clearly summarize the results of the paper. Overall, the main findings of the paper need to be more clearly communicated. 8. The hypothesis should be clearly stated at the end of the introduction.